# A comparison of respiratory particle emission rates at rest and while speaking or exercising

Christopher M. Orton [1,2,3,12 ✉], Henry E. Symons [4,12], Benjamin Moseley[1], Justice Archer [4], Natalie A. Watson [5], Keir E. J. Philip [1,3], Sadiyah Sheikh[4], Brian Saccente-Kennedy[6], Declan Costello[7], William J. Browne[8], James D. Calder[9,10], Bryan R. Bzdek[4], James H. Hull[1,11], Jonathan P. Reid [4 ✉] & Pallav L. Shah [1,2,3 ✉]

## Abstract

**Background** The coronavirus disease-19 (COVID-19) pandemic led to the prohibition of group-based exercise and the cancellation of sporting events. Evaluation of respiratory aerosol emissions is necessary to quantify exercise-related transmission risk and inform mitigation strategies.

**Methods** Aerosol mass emission rates are calculated from concurrent aerosol and ventilation data, enabling absolute comparison. An aerodynamic particle sizer (0.54–20 µm diameter) samples exhalate from within a cardiopulmonary exercise testing mask, at rest, while speaking and during cycle ergometer-based exercise. Exercise challenge testing is performed to replicate typical gym-based exercise and very vigorous exercise, as determined by a preceding maximally exhaustive exercise test.

**Results** We present data from 25 healthy participants (13 males, 12 females; 36.4 years). The size of aerosol particles generated at rest and during exercise is similar (unimodal ~0.57–0.71 µm), whereas vocalization also generated aerosol particles of larger size (i.e. was bimodal ~0.69 and ~1.74 µm). The aerosol mass emission rate during speaking (0.092 ng s$^{-1}$; minute ventilation (VE) 15.1 L min$^{-1}$) and vigorous exercise (0.207 ng s$^{-1}$, $p = 0.726$; VE 62.6 L min$^{-1}$) is similar, but lower than during very vigorous exercise (0.682 ng s$^{-1}$, $p < 0.001$; VE 113.6 L min$^{-1}$).

**Conclusions** Vocalisation drives greater aerosol mass emission rates, compared to breathing at rest. Aerosol mass emission rates in exercise rise with intensity. Aerosol mass emission rates during vigorous exercise are no different from speaking at a conversational level. Mitigation strategies for airborne pathogens for non-exercise-based social interactions incorporating vocalisation, may be suitable for the majority of exercise settings. However, the use of facemasks when exercising may be less effective, given the smaller size of particles produced.

## Long summary

SARS-CoV-2, the virus that causes COVID-19, and other respiratory viruses are transmitted via respiratory particles emitted while breathing or speaking. Transmission of these viruses will depend in part on the rate at which these particles are emitted. Here, we studied respiratory particle sizes and emission rates in healthy people while breathing at rest, while speaking and during exercise on a static bicycle. We find that speaking generates larger particles and exercise generates smaller particles. The particle emission rate during speaking and typical gym-based exercise was similar but lower than values measured during very vigorous exercise. These findings help us to understand the emission of respiratory particles during different activities, and suggest that preventative measures for COVID-19 such as social distancing, used for non-exercise-based social interactions involving speaking, may be suitable for the majority of exercise settings.

[1] Department of Respiratory Medicine, Royal Brompton Hospital, London, UK. [2] Department of Respiratory Medicine, Chelsea & Westminster Hospital, London, UK. [3] National Heart and Lung Institute, Guy Scadding Building, Imperial College London, Dovehouse Street, London, UK. [4] School of Chemistry, University of Bristol, Bristol, UK. [5] Department of Ear, Nose and Throat Surgery, Guys & St. Thomas NHS Foundation Trust, London, UK. [6] Speech and Language Therapy Department, Royal National Ear Nose and Throat Hospital, London, UK. [7] Ear, Nose and Throat Department, Wexham Park Hospital, Slough, UK. [8] School of Education, University of Bristol, Bristol, UK. [9] Department of Bioengineering, Imperial College London, London, UK. [10] Fortius Clinic, Fitzhardinge St, London, UK. [11] Institute of Sport, Exercise and Health (ISEH), UCL, London, UK. [12] These authors contributed equally: Christopher M. Orton, Henry E. Symons. ✉email: c.orton@rbht.nhs.uk; j.p.reid@bristol.ac.uk; pallav.shah@imperial.ac.uk

The global coronavirus disease-19 (COVID-19) pandemic, caused by severe acute respiratory syndrome coronavirus 2 (SARS-CoV-2), has led to one of the most significant public health emergencies of the last century, placing unprecedented challenges on healthcare systems worldwide[1,2]. Public health measures aimed at reducing rates of infection have focussed on social distancing and the use of face coverings. In addition, the cancellation and prohibition of social and cultural events have resulted in marked societal disruption, impacting a plethora of sporting occasions, including the 2020 Olympic Games[3–8]. Concern regarding SARS-CoV-2 transmission during exercise also resulted in the temporary closure of many indoor exercise facilities and curtailed access to sporting and physical group activities[9–14].

Despite this, it is well recognised that exercise is essential to health and wellbeing. Accordingly, the World Health Organisation recommends >150 min of moderate-intensity or 75 min of vigorous intensity physical activity per week[15]. Regular physical activity is associated with improved quality of life, reduced cardiovascular risk and mental health benefits[16–18]. In addition to individual benefits, the sport and physical activity leisure industry is a substantial employer, contributing an estimated $756 billion to the global economy[19]. It is also recognised that regular exercise and improved physical activity status may mitigate the risk of severe COVID-19[20].

Respiratory disease transmission is governed by host, recipient, pathogen and environmental factors[21,22]. SARS-CoV-2 transmission occurs predominantly through respired particles containing the virus[23,24]. Such particulate matter is expelled during exhalation, with an arbitrary distinction made between aerosols and droplets, as particles smaller and larger than 5 μm diameter, respectively[25,26]. Recently, it has been recognised that a more correct distinction should be made separating particles below and above 100 μm, with all particles smaller than 100 μm inhalable and exhibiting similar aerodynamic behaviour with respect to suspension in air[27]. We previously demonstrated that the number and mass concentrations of aerosols released during breathing and vocalising are directly related to the intensity and type of activity performed and that vocalising produces larger particles than breathing, consistent with studies by other researchers[26,28–30]. The intensity-dependence of aerosol generation is likely relevant during sporting activity, given the hyperpnoea associated with vigorous physical activity[31]. However, despite their importance for transmission modelling and evidence-based mitigation interventions, absolute measurements of respiratory aerosol generation during exercise have only been studied in a limited capacity[32–34]. Performing such measurements in an environment with zero aerosol background concentration levels is essential to ensure that every detected exhaled particle arises from respiratory activity, rather than from inhaled particle-laden air[33].

Vaccination, social distancing and other factors have reduced SARS-CoV-2 transmission in certain contexts. However, an improved understanding of airborne viral transmission and mitigation strategies remains relevant to all respiratory pathogens, including SARS-CoV-2. This study investigates the generation of respiratory aerosol during exercise, comparing these emissions to those produced by breathing at rest and by speaking at a conversational level, both activities upon which current transmission mitigation guidance are based. We hypothesised that exercise would generate more aerosol than at rest or during the conversational speech, due to the increased ventilation rates and effort associated with exercise.

We demonstrate that the size of aerosol particles generated at rest and during exercise is similar, whereas vocalisation also generates aerosol particles of a larger size. The aerosol mass emission rate during speaking and vigorous exercise is similar, but lower than values measured during very vigorous exercise. Mitigation strategies for airborne pathogens, deemed appropriate for non-exercise-based social interactions incorporating vocalisation, may be suitable for the majority of exercise settings.

## Methods

**Study design and participants**. "The Investigation of ParticulatE Respiratory Matter Release During perFormance and Exercise to infOrm Guidance in the SARS-CoV-2 PandeMic" (PERFORM-2) is an observational study investigating respiratory aerosol generation during a range of activities impacted by the COVID-19 pandemic, including singing and playing musical instruments[28,29]. In this study, 25 healthy individuals, who were free from significant cardiovascular or respiratory illness, were recruited to include a range of athletic abilities across both sexes.

Participants attended on a single occasion for testing. The participants exhibited no COVID-19 symptoms, were lateral flow negative, and refrained from vigorous exercise, smoking, consuming alcohol, or eating heavily for 4 h prior to testing[35]. Written informed consent was provided by all participants and study approval was granted by the Public Health England Research Ethics and Governance of Public Health Practice Group (PHE REGG, NR0221). Written informed consent for publication of the images depicting the experimental setup was obtained. The sample size of 25 participants was chosen to allow detection of differences in the magnitude of 2.5 with a power of 0.9, using sampling variances from a previous study (PERFORM-1) in calculations and allowing for a potential dropout rate of 10%[28].

**Study protocol**. The study protocol is illustrated conceptually in Fig. 1b. Following familiarisation with the experimental setup, participants underwent a maximal cardiopulmonary exercise testing (CPET) on a cycle-ergometer [CORTEX MetaLyzer 3B-R3 + Wattbike Atom (Next Generation) cycle-ergometer or Vyaire™ Medical Vyntus CPX + VIAsprint 200 P W/BP Serial Ergometer system] to voluntary exhaustion, per CPET international guidelines, to characterise exercise capacity and ventilatory response[35].

Following at least 1 h of rest, participants then completed a second exercise test where both ventilatory and aerosol measurements (see detail below) were concurrently evaluated using an adapted facemask (see Fig. 1a). First, ventilation and aerosol generation at rest was measured over 1 min. Then, ventilatory and aerosol measurements were made whilst participants vocalised a set text at a constant pace of at least 70 dBA (A-weighted decibels). The sound level was measured by a sound level metre placed 30 cm from the mouth and 70 dBA was selected as a target for participants, to enable comparison with previous measurements of conversational speaking in the 70–80 dBA range, while also accounting for attenuation attributable to the CPET mask (see Supplementary Figures 5 and 6)[28,36,37]. Next, participants were instructed to begin exercising, with two fixed periods of constant work, prescribed from the CPET and selected to replicate work intensities of vigorous intensity (80% of the anaerobic workload, for ~6 mins) and very vigorous (anaerobic workload +30% (maximal workload minus anaerobic workload), for ~4 mins) gym-based exercise (see Fig. 1b). Aerosol measurements were taken following 2 min of vigorous exercise and following 30 sec of very vigorous exercise[38]. Perceived exertion was assessed at rest, during vigorous exercise and during very vigorous exercise using the BORG CR-10 Scale[39]. CPET data were analysed using Cortex MetaSoft® Studio Version 5.12.0 (Cortex system) and SentrySuite® software V. 320 (Vyaire™ Medical system).

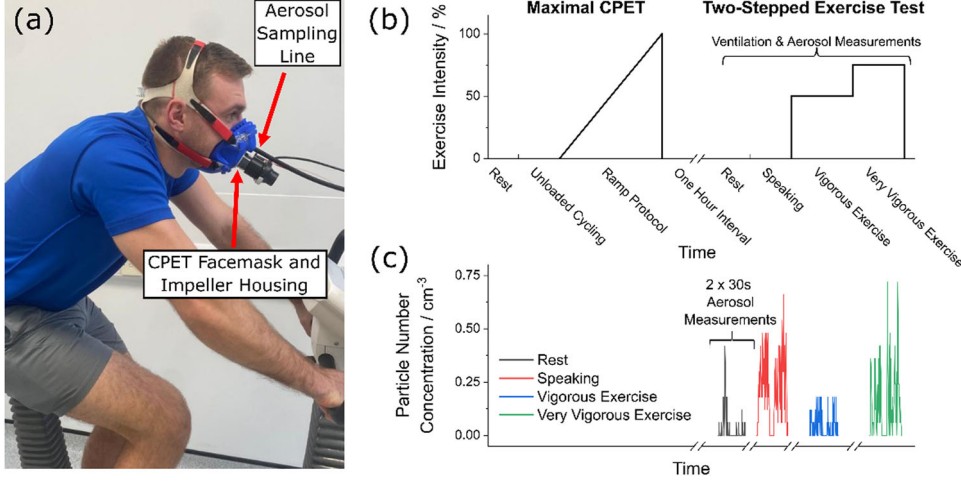

**Fig. 1 Experimental setup and study protocol. a** Experimental setup: cycle-ergometer-based exercise testing with concurrent ventilatory and aerosol measurements via APS-mask, from sampling line inserted into sampling hole at the tip of the nose. Informed consent for publication of the image was obtained. **b** Experimental design: maximal CPET followed by a two-stepped exercise test, post 1-h interval. **c** Typical time series of aerosol number concentration data sampled periodically via APS-mask during the two-stepped exercise test. Horizontal scale adjusted to align with corresponding activities in **b**. *APS* aerodynamic particle sizer, *CPET* Cardiopulmonary exercise testing.

**Aerosol measurements (0.54–20 μm diameter).** Aerosol measurements were accomplished using an aerodynamic particle sizer (APS; TSI Inc. model 3321; 1 L min$^{-1}$ sample flow rate, 4 L min$^{-1}$ sheath flow rate, 1-second sampling interval), which measures the number concentration and size distribution of aerosol particles in the 0.54–20 μm diameter size range. The APS size range overlaps with that associated with the vast majority of respiratory aerosol by number, allowing definitive characterisation of the size distributions typically ascribed as the bronchiolar and laryngeal modes[26]. This study was conducted in a laminar flow operating theatre with sufficient air changes per hour within the ultraclean ventilation canopy to ensure the background aerosol number concentration within the APS size range was 0 particles cm$^{-3}$. Consequently, aerosol detected by the APS can be confidently attributed to the participant, with background aerosol concentrations returning to 0 cm$^{-3}$ during sampling pauses. Room temperature was controlled at 18 °C, with relative humidity (RH) 40%.

Aerosol measurements were taken in two sampling configurations. In the main configuration (see Fig. 1a), aerosol was sampled from a modified CPET facemask (referred to as APS-mask). The silicone mask (Hans Rudolph 7450 Series V2) was adapted with a 6 mm sampling hole cut at the tip of the nose to facilitate the passage of the aerosol sampling line (while sampling aerosol in this configuration) or with a tight-fitting silicone bung (while performing ventilatory measurements alone). The sampling port site was selected to avoid the collection of water droplets pooling in the facemask. In the second sampling configuration, aerosol was sampled without a CPET mask present using a 3D-printed funnel positioned 15–20 cm from the participant's mouth (referred to as APS-cone; the sampling funnel is visible in Supplementary Figure 1). As demonstrated by the example time series in Fig. 1c, all APS-mask and APS-cone measurements were made during two, 30-second periods unless the participant struggled to maintain the activity, in which case the measurement period was shorter. The data presented are based on the mean values for each 30-second sampling period. Results reported in the main body of this manuscript were all taken in the APS-mask configuration, as this configuration was the more robust aerosol

sampling approach. Supplementary Methods 1 provides detailed validation of this aerosol sampling methodology.

Aerosol number concentrations and size distributions were extracted directly from the time-averaged APS data. APS data were analysed using the Aerosol Instrument Manager v10.3 (TSI inc) software. Aerosol mass concentrations were calculated based on the mean diameter of each size bin. A particle density of 1 g cm$^{-3}$ is assumed, as aerosol is generated in the respiratory tract at very high RH (>99%)[40,41]. Additionally, our recent work based on analysis of the sampling of aerosols through the collection funnel and into the APS shows that the full-size distributions (0.54–20 μm) reported here can fully equilibrate in size to the sampling RH, with sufficient time from exhalation to size measurement by the APS instrument[42]. Although our previous study suggests the RH for the measured size distributions remains high, we cannot unambiguously state the RH at which our size distributions are measured and this will be the subject of a future study. A comparison in terms of aerosol mass concentration assumes the potential dose transmitted by an infected individual scales with particle volume.

A key aspect of this study is the concurrent measurement of aerosol concentration and ventilation, with the APS sampling directly from the CPET facemask. The aerosol measurements allow quantification of the aerosol concentration in the expiratory jet. The ventilation measurements permit quantification of the total flow rate of the expiratory jet (see Table 1). When combined, the separate aerosol and ventilation measurements enable estimation of the absolute number of particles and amount of aerosol emitted during each activity, per unit time, which are the absolute aerosol number and mass emission rates, respectively. Estimates of emission rates are important to achieve as they enable absolute comparisons across activities with differing rates of ventilation of the number and mass of respiratory particles emitted by a participant per second.

Aerosol emission rates were estimated based on the synchronous measurements of both aerosol concentrations and VE for the four activities investigated. Due to the concurrent nature of these measurements, these values were determined independently for both repeats of each of the four activities, prior to any statistical analysis.

**Table 1 Demographic and exercise physiology data.**

| | | Variables | Mean ± SD |
|---|---|---|---|
| Demographics | | Sex | 13 males 12 females |
| | | Age/years | 36.4 ± 14.9 |
| | | BMI | 23.8 ± 4.1 |
| Maximal CPET | | Maximal work rate/W | 304 ± 78 |
| | | HR %max/% | 96% ± 0.04 |
| | | VE/L min$^{-1}$ | 120.13 ± 45.83 |
| | | BF/bpm | 48 ± 12 |
| | | VT/L | 2.48 ± 0.68 |
| | | VO$_2$ max% predicted/% | 130 ± 26 |
| | | VO$_2$/kg/ml kg$^{-1}$ min$^{-1}$ | 42.4 ± 11.01 |
| | | RER | 1.24 ± 0.1 |
| Two-stepped exercise test | Rest | HR %max | 43 ± 0.05 |
| | | VE/L min$^{-1}$ | 11.44 ± 3.93 |
| | | BF/bpm | 15 ± 4 |
| | | VT/L | 0.85 ± 0.4 |
| | | VO$_2$ max% predicted/% | 16.3 ± 5.1 |
| | | RER | 0.84 ± 0.07 |
| | | BORG CR-10 scale | "Nothing at all" 0.06 ± 0.22 |
| | Speaking | HR %max | 45 ± 0.06 |
| | | VE/L min$^{-1}$ | 15.1 ± 5.58 |
| | | BF/bpm | 23 ± 5 |
| | | VT/L | 0.93 ± 0.32 |
| | Vigorous exercise | Work rate/W | 152 ± 52 |
| | | HR %max/% | 79 ± 0.06 |
| | | VE/L min$^{-1}$ | 62.62 ± 17.94 |
| | | BF/bpm | 29 ± 5 |
| | | VT/L | 2.22 ± 0.64 |
| | | VO2 max% predicted/% | 97.9 ± 22.8 |
| | | RER | 0.94 ± 0.05 |
| | | BORG CR-10 Scale | "Somewhat strong" 4.16 ± 1.37 |
| | Very vigorous exercise | Work rate/W | 226 ± 66 |
| | | HR %max | 92 ± 0.05 |
| | | VE/L min$^{-1}$ | 113.61 ± 38.73 |
| | | BF/bpm | 47 ± 12 |
| | | VT/L | 2.46 ± 0.62 |
| | | VO2 max% predicted/% | 126.9 ± 24.1 |
| | | RER | 1.06 ± 0.04 |
| | | BORG CR-10 scale | "Very strong" 8.46 ± 1.78 |

$n = 25$ participants. Maximal CPET values are those achieved maximally. Body mass index (BMI), oxygen uptake (VO$_2$ kg$^{-1}$), heart rate (HR), minute ventilation (VE), breathing frequency (BF), tidal volume (VT), respiratory exchange ratio (RER). VO$_2$ max% predicted/% calculated for rest, vigorous, and very vigorous exercise relative to maximal CPET[39].

The aerosol number emission rates are calculated by Eq. (1):

$$\text{Number emission rate } (s^{-1})$$
$$= \frac{\text{Number concentration}(L^{-1}) \times \text{Ventilation}(L\,min^{-1})}{60}$$

The mass emission rates are calculated by Eq. (2):

$$\text{Mass emission rate } (ng\,s^{-1})$$
$$= \frac{\text{Mass concentration}(ng\,L^{-1}) \times \text{Ventilation}(L\,min^{-1})}{60}$$

**Statistical analysis**. The aerosol data are clustered by activity: for each participant there are eight sets of measurements representing the four different activities (rest, speaking at 70–80 dBA, vigorous exercise and very vigorous exercise) and two replications of each per participant. For each response variable considered (number concentration, mass concentration, number emission rate and mass emission rate) a positively skewed histogram was observed, with log transformation producing an approximately symmetric distribution that can be represented by a normal distribution. Note that aerosol generation and aerosol size distributions are lognormally distributed across participants for all

activities, consistent with the previous studies[25,28,29]. Given the clustering, a two-level random effects model was used to fit each (logged) response variable (with measures nested within participants) using MLwiN v3.05[43]; fixed effects were included for activity type, to be able to compare activity types while adjusting for participant differences. Initially, activity type was assessed using a likelihood ratio test and then each pair of activities was compared using (two-sided) Wald tests to identify which activities significantly differ. Missing data were assumed to have occurred at random.

**Reporting summary**. Further information on research design is available in the Nature Research Reporting Summary linked to this article.

## Results

**Subject characteristics and exercise capacity**. Twenty-five healthy participants (13 males, 12 females), with a mean age of 36.4 years, (SD ± 14.9 years, range 19–72) and normal body mass index (BMI) at 23.8 kg m$^{-2}$ (SD ± 4.1), completed a two-component CPET protocol on a cycle-ergometer, as shown in Fig. 1a. The protocol is summarised schematically in Fig. 1b (see Methods). Participants initially completed a maximal CPET,

exhibiting a broad range of athletic capability, with a mean peak oxygen uptake per kg ($VO_2$ $kg^{-1}$) of 42.4 ml $kg^{-1}$ $min^{-1}$ (SD ± 11.01, range 26 to 65). A subsequent two-stepped CPET was used to determine physiological parameters (see Table 1) during four discrete activities: breathing at rest, speaking at a conversational volume of 70–80 dBA, and breathing during vigorous and very vigorous exercise. The work rate intensity of the latter two activities was derived from each participant's maximal CPET data (see Methods). The intensity of exercise was subjectively assessed by the participants as "Somewhat strong" and "Very strong" work, based on mean BORG CR-10 Scale during vigorous exercise and very vigorous exercise, respectively (see Table 1)[39]. Mean minute ventilation (VE) for breathing at rest, speaking, vigorous exercise and very vigorous exercise were 11.4 (SD ± 3.9), 15.1 (SD ± 5.6), 62.6 (SD ± 17.9) and 113.6 (SD ± 38.7) L $min^{-1}$, respectively (see Table 1).

During the two-stepped exercise test protocol, aerosol number concentrations were measured via a modified CPET facemask with the sampling line to an APS inserted at the tip of the nose, as shown in Fig. 1a. Mass concentrations were inferred from the number and diameter of the particles detected (see Methods). This setup, referred to as APS-mask, was used to record aerosol concentrations during two 30-second sampling periods for each of the four activities described above, with typical data obtained from one participant shown in Fig. 1c. Complete aerosol measurement data were collected from 20 participants, with five participants unable to complete the second repeat measurement during very vigorous exercise due to exhaustion. In addition, the aerosol concentration from one replication of one participant's breathing measurements was identified as erroneously high, likely due to systematic error, and was therefore excluded from subsequent analysis. Figure 2 summarises the time-averaged recorded aerosol number concentrations (particles $cm^{-3}$) and mass concentrations ($\mu g\,m^{-3}$) in the 0.54–20 µm diameter size range from the exhalatory jet, across all participants. Following adjustment for multiple comparisons, no significant differences were identified between male and females participants (Supplemental Figure 8).

Vigorous exercise and very vigorous exercise generate approximately four and eight times greater numbers of aerosol particles (based on median values) than breathing at rest, respectively ($p < 0.001$) (see Fig. 2a and Table 2). Moreover, approximately twice as many aerosol particles are generated during very vigorous exercise than during vigorous exercise ($p < 0.001$). Speaking at 70–80 dBA generates more aerosol particles than vigorous exercise ($p < 0.001$) but a similar amount to very vigorous exercise ($p = 0.92$).

The aerosol mass concentration during vigorous exercise is nearly 9 times higher than resting breathing ($p < 0.001$), whereas very vigorous exercise generates an aerosol mass concentration more than 20 times higher than resting breathing ($p < 0.001$) (see Fig. 2b, Table 2 and Supplementary Table 1). Speaking generates a higher mass concentration than vigorous exercise ($p < 0.001$) but a similar mass concentration to very vigorous exercise ($p = 0.083$).

**Aerosol size distributions**. Figure 3 reports mean aerosol size distributions for all 25 participants, reported both on logarithmic and linear scales for clarity. Corresponding mean volume size distributions and cumulative volume fractions are reported in Supplementary Figure 4. The aerosol size distributions for breathing at rest and during vigorous and very vigorous exercise are similar in shape, all well-described by a single-mode with maximum number concentrations at 0.57 (SE ± 0.04), 0.59 (SE ± 0.06) and 0.71 (SE ± 0.02) µm diameter, respectively.

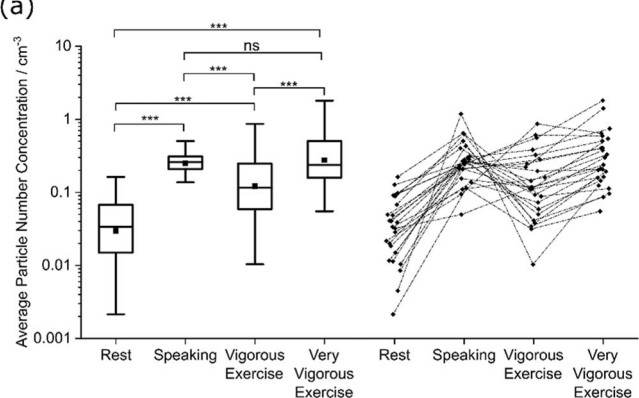

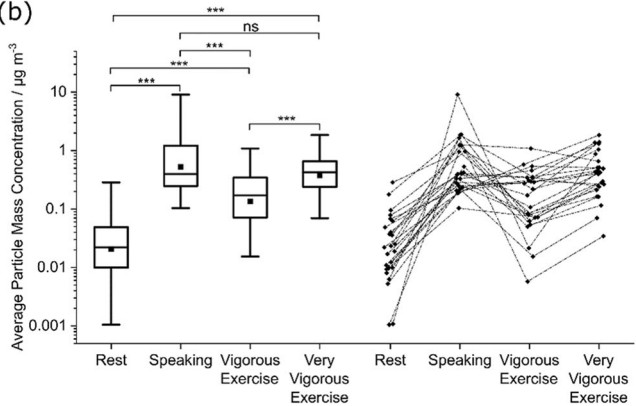

**Fig. 2 Aerosol number and mass concentrations for breathing at rest and during exercise, and while speaking.** Box and whisker plots (left) and corresponding data (right) showing average particle number concentrations (**a**) and mass concentrations (**b**) for the same series of activities (breathing at rest, during vigorous exercise, during very vigorous exercise, and speaking at 70–80 dBA) across $n = 25$ participants, sampled via APS-mask. Boxes indicate mean, median and IQR, whiskers indicate range (data within 1.5 IQR), *** indicates $p < 0.001$, not significant (ns). *ns* not significant, *APS* aerodynamic particle size, *IQR* interquartile range.

Concentrations for all three activities decrease sharply at aerodynamic diameters >1 µm. Conversely, the average size distribution generated by speaking at 70–80 dBA displays two modes: one (similar to breathing and exercise) with a maximum at 0.69 (SE ± 0.01) µm diameter and a second at 1.74 (SE ± 0.10) µm diameter. Full fitting parameters for all activities are provided in Supplementary Table 2.

**Aerosol number and mass emission rates**. Median aerosol number emission rates for vigorous and very vigorous exercise (see Fig. 4a, Table 2, Supplementary Table 3) are both substantially higher than the number emission rates for resting breathing and speaking ($p < 0.001$). In terms of aerosol mass emission rate (see Fig. 4b, Table 2), very vigorous exercise generates a median value of 0.682 ng $s^{-1}$ (IQR 0.31–1.28), a value greater than during rest, speaking or vigorous exercise ($p < 0.001$). However, there is no difference between the aerosol mass emission rate during vigorous exercise (median 0.207 ng $s^{-1}$, IQR 0.053–0.36) and speaking (median 0.092 ng $s^{-1}$, IQR 0.061–0.23; $p = 0.726$).

## Discussion

This study characterises aerosol generation (diameter 0.54–20 µm) in healthy subjects, during exercise, speaking at a conversational

**Table 2 Summary of aerosol concentration, emission rate and size distribution data.**

| Activity | Aerosol number concentration/cm⁻³ (IQR) | Aerosol Mass Concentration/μg m⁻³ (IQR) | Aerosol number emission rate/s⁻¹ (IQR) | Aerosol mass emission rate/ng s⁻¹ (IQR) | Aerosol modal diameter/Dₚ/μm (SE) |
|---|---|---|---|---|---|
| Rest | 0.03 (0.01−0.07) | 0.02 (0.01−0.07) | 6.5 (2−14) | 0.003 (0.001−0.01) | 0.57 (±0.04) |
| Speaking | 0.26 (0.21−0.31) | 0.40 (0.25−1.22) | 58.5 (43−98) | 0.092 (0.06−0.23) | 0.69 (±0.01) and 1.74 (±0.10) |
| Vigorous exercise | 0.12 (0.06−0.25) | 0.17 (0.06−0.25) | 145 (46−285) | 0.207 (0.05−0.36) | 0.59 (±0.06) |
| Very vigorous exercise | 0.24 (0.16−0.50) | 0.42 (0.24−0.66) | 625 (230−1003) | 0.682 (0.31−1.28) | 0.71 (±0.02) |

Median aerosol number and mass concentrations and emission rates and size distribution modes were obtained from breathing at rest, speaking, vigorous exercise and very vigorous exercise across n = 25 participants.

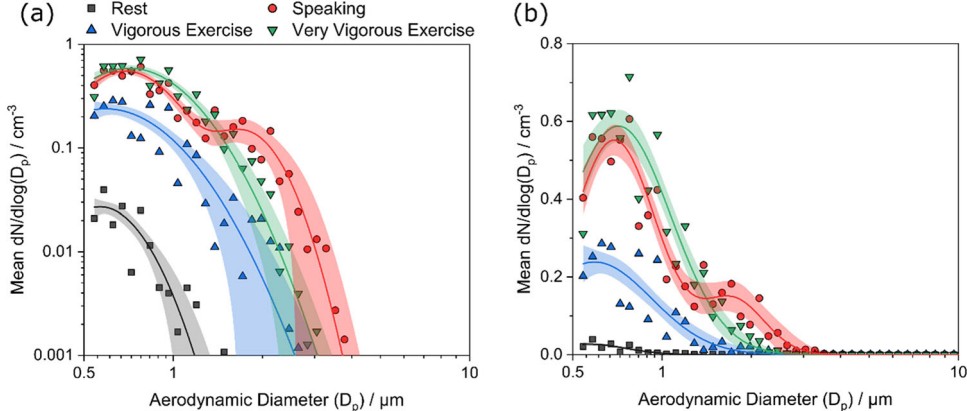

**Fig. 3 Aerosol size distributions (0.54–20 μm) for breathing at rest and during exercise, and while speaking.** Comparison of average size distributions from speaking at 70–80 dBA (red circles), breathing at rest (black squares), and during vigorous (blue triangles) and very vigorous (green triangles) exercise across n = 25 participants. Corresponding log-normal fits for each activity are shown by curves of the same colour, with shaded areas representing the 95% confidence interval of the fit. Data depicted on log base 10 (**a**) and linear (**b**) scales. dBA A-weighted decibels.

level and breathing at rest. The key findings are that the aerosol size distribution generated during exercise is consistent with that generated whilst breathing at rest and that aerosol mass emission rates increase with the hyperpnoea of increasing exercise intensity. Furthermore, there is no difference between the aerosol mass emission rates during vigorous exercise and during speaking at a conversational level, i.e., being in proximity to an individual undertaking gym-based exercise poses no higher risk of aerosol-based transmission of respiratory pathogens than being in close proximity during a conversation.

Aerosol number concentrations describe the quantity of aerosol particles released within the 0.54–20 μm size range. Aerosol mass concentrations assess the volume of the material generated within the 0.54–20 μm size range, as inferred by particle diameter and assuming all particles have a density of 1 g/cm³, equivalent to water. Aerosol number and mass concentrations increased with exercise intensity (see Fig. 2), demonstrating an effort-related dependence equivalent to that previously found during speaking and singing[25,28]. Aerosol number and mass concentrations while speaking at a conversational level (70–80 dBA) are greater than those generated during vigorous exercise but no different to those generated during very vigorous exercise. This finding indicates that in terms of raw aerosol particle measurements (not normalised for ventilation), release during very vigorous exercise is equivalent to vocalisation.

Analysis of aerosol size distributions for breathing at rest, during vigorous exercise and during very vigorous exercise indicates that these activities produce an expiratory output with similar particle size. Specifically, these three activities displayed unimodal distributions of particles centred at median sizes of 0.57 (SE ± 0.04), 0.59 (SE ± 0.06) and 0.71 (SE ± 0.02) μm diameter, respectively (see Fig. 3, Table 2). In contrast, the aerosol size distribution generated during speaking is bimodal, consisting of

one mode at approximately 0.69 (SE ± 0.01) μm diameter (equivalent to the breathing mode) and a second mode at 1.74 (SE ± 0.10) μm diameter attributed to vocalisation. This finding is consistent with previous studies demonstrating that vocalisation generates aerosol at larger particle sizes than breathing alone[25,26,28]. The larger particles arising from vocalisation are likely to be generated in the larynx and created by the movement of secretions arising from glottic closure, whereas smaller particles are more likely to have arisen from the distal airway tract and small airways during respiration[26]. The volume size distribution (see Supplementary Figure 4) clearly shows an enhanced laryngeal mode with most of the volume (mass) of the particles generated having aerodynamic sizes below 5 μm. The finding that the size distribution and fraction of aerosol particles generated during exercise matches that generated during breathing at rest are relevant from a public health perspective. The majority of face coverings and commercially available facemasks (i.e., a cloth or surgical facemask) are less efficient at removing particles in this size range (0.3–1 μm diameter), increasing in efficiency at removing larger sizes, such as those produced in higher concentrations during vocalisation[44]. More specifically, a recent study demonstrated decreased efficiency of outward protection with commonly used facemasks from particles of 0.7 μm diameter (~65% for surgical, and ~25% for cloth), to those of 2 μm diameter (~75% efficiency for surgical, and ~50% for cloth masks)[45]. A recommendation to wear face coverings during exercise in order to reduce apparent aerosol transmission risks, is therefore likely to be less effective than for activities that involve vocalisation, such as singing. However, wearing high-grade face coverings will still reduce aerosol emissions.

Raw aerosol number and mass concentrations characterise the aerosol particles sampled within a given volume of exhaled air but comparing these figures directly would assume that ventilation

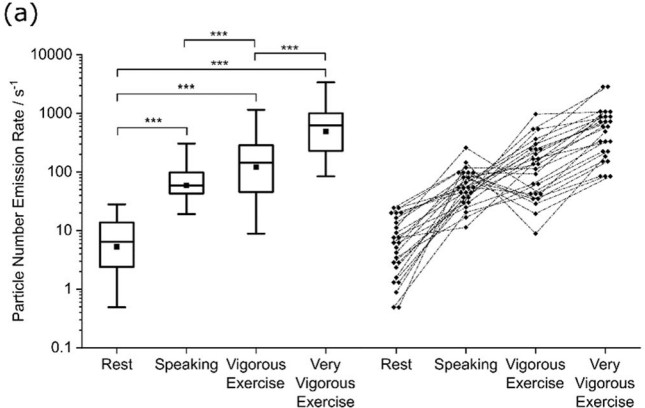

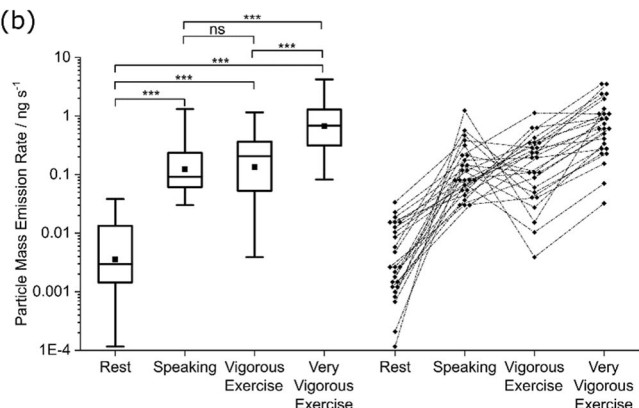

**Fig. 4 Aerosol number and mass emission rates for breathing at rest and during exercise, and while speaking.** Box and whisker plots (left) and corresponding data (right) showing particle number emission rates (**a**) and mass emission rates (**b**) for the same series of activities (breathing at rest, during vigorous exercise, during very vigorous exercise and speaking at 70–80 dBA) across $n = 25$ participants. Boxes indicate mean, median and IQR, whiskers indicate the range (data within 1.5 IQR), *** indicates $p < 0.001$, not significant (ns). *ns* not significant, *dBA* A-weighted decibels; *IQR* interquartile range.

rates are consistent across differing activities. To maximise clinical utility, it is therefore important to consider the markedly different rates of ventilation, which ranged from a mean 11.4 to 113.6 L min$^{-1}$ across the activities (rest to very vigorous exercise) investigated in this study. In combination, these synchronous measurements of aerosol number and mass concentration, and measurements of ventilation, enable estimates of absolute total aerosol particle number and mass emission rates to be determined.

Whilst speaking generated greater raw number concentrations than vigorous exercise ($p < 0.001$) and similar concentrations to very vigorous exercise ($p = 0.92$), the ventilation rates for vigorous and very vigorous exercise were greater than during speaking at a conversational level (mean 62.6, 113.6 and 15.1 L min$^{-1}$, respectively) resulting in higher aerosol number emission rates for vigorous and very vigorous exercise, than those for speaking ($p < 0.001$). However, whereas very vigorous exercise also generated a higher mass emission rate than speaking ($p < 0.001$), importantly, the mass emission rate for vigorous exercise was no different from speaking (p=0.726). This higher mass emission rate, compared to number emission rate is a direct consequence of the additional mode of larger particles generated by vocalisation (see Supplementary Figure 4), which contribute disproportionately as larger particles carry greater mass.

Absolute aerosol emission rates have important clinical utility as they quantify the total number and mass of aerosol particles released by an individual per second, and provide an absolute basis upon which to compare aerosol particle release between individuals or from different activities. In addition, the findings enable computational modelling and calculation of transmission risk to be performed more accurately, along with enhanced ventilation planning (i.e., for gyms and indoor exercise facilities): aerosol accumulation in any indoor space can be deduced if the rates of aerosol emission and building ventilation are known. Furthermore, the absolute quantity of potentially infective material produced on exhalation can be estimated from aerosol mass emission rates and viral load, if known. However, the variation in viral load with particle size remains uncertain.

The finding that aerosol mass emission rates during vigorous exercise are not different to release during speaking at a conversational level has broad applicability to the safe practice of sports and exercise worldwide. Mitigation strategies deemed appropriate for speech and breathing at rest in the context of a pandemic environment may be suitable for aerosol particle release during the majority of exercise settings, i.e., vigorous exercise intensity or less. Although aerosol mass emission rates during very vigorous exercise, were approximately 3.5 times greater than during vigorous exercise, and were significantly higher than during conversational speech, such high-intensity exercise is by its nature unsustainable. Our previous report that vocalisation above conversational volume at 90–100 dBA generates an aerosol mass concentration 14 times greater than speaking at a conversational volume, suggests that aerosol release from exercise can be considered in a similar manner as vocalisation in a real world setting. Bursts of very vigorous exercise or loud vocalisation, such as in a public house or bar lead to an increased risk of transmission.

Although aerosol emission rates are important to quantify, we recognise that they are only one part of a complex interplay of factors governing infective transmission between individuals. Other factors including viral load, recipient inhalatory rate (which is activity and intensity-dependent) and environmental factors such as temperature, humidity, ventilation rates and the wearing of face coverings all influence transmission of airborne pathogens. VE rates during exercise are higher than during speech, which could increase the quantity of potentially infectious aerosol particles inhaled by a recipient.

Number and mass concentrations recorded when speaking at 70–80 dBA are consistent with measurements in our previous study, validating the sampling methodology used in this study (see Supplementary methods 1)[28]. Simultaneous measurement of aerosol and ventilatory parameters required sampling directly out of the facemask, which did not impact the ventilatory data recorded (see Supplementary Figure 7). In fact, this approach was a more robust aerosol sampling configuration than sampling via a funnel positioned distal to the participant's mouth, as participant movement during exercise led to wide variations in the position of the participant's face (the aerosol source) relative to the sampling funnel. Consequently, aerosol concentrations measured through the funnel were lower than those measured in previous work. It is likely that the large distance between the participant and the funnel is a key factor in the reduced particle concentration recorded. Similar number concentrations, mass concentrations and size distributions to those measured in previous work were sampled directly through the facemask (see Supplementary methods 1, Supplementary Figures 2 and 3)[28]. Whilst previously characterised as constituting a small proportion of the total mass of exhaled material, particles larger than 20 μm in diameter were not assessed in this study[26]. Additionally, the value of the measurements that were made in this study are supported by the

recent finding that higher levels of the SARS-CoV-2 virus are carried by the fine aerosol particles (<5 μm diameter) produced by infected individuals[46].

The cohort of participants included in the study covers a broad range of ages and athletic abilities, with near equal division of the sexes, however, we note that individuals with differing cardio-pulmonary (and other) disorders were not included in the work and may behave differently. The exercise work rate thresholds selected for the study represent activity levels that an individual would perform in the gym for a sustained period (vigorous exercise) and for short intervals (very vigorous exercise), with five participants unable to complete the period of very vigorous workload due to exhaustion. Importantly, given that the workload thresholds for each individual's second exercise test were determined by their performance in their first test, both the vigorous exercise and very vigorous exercise workloads were physiologically relevant to the individual performing the test. The appropriate selection of these intensities was corroborated by the heart rate, RER and BORG CR-10 Scale data presented, with participants assessing the vigorous exercise and very vigorous exercise as "Somewhat strong" and "Very strong", respectively[39].

In conclusion, the size distribution of aerosol generated during exercise is consistent with that generated during breathing and is dominated by particles <1 μm in diameter, whereas vocalisation produces additional larger particles. Estimates of absolute aerosol number and aerosol mass emission rates are presented, derived from concurrent measurements of aerosol concentrations and ventilation rates. The aerosol mass emission rate from an individual during most exercise (i.e., of vigorous intensity or less) is equivalent to, or less than, that emitted by the same individual while speaking at a conversational level. Moreover, when considering these findings in the context of previous work, the authors propose that punctuating bursts of greater emissions during non-sustainable very vigorous exercise may be considered in the same manner as increasing speech loudness above a background conversational level. These findings have broad applicability to the safe practice of sports and exercise worldwide in the context of a pandemic environment, as mitigation strategies deemed appropriate for social situations may be suitable for the majority of exercise settings.

## Data availability

Source data underlying the figures and the raw data used in the analysis have been made publicly available in the BioStudies database, https://www.ebi.ac.uk/biostudies/, under accession ID S-BSST691.

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

## Acknowledgements
The authors acknowledge funding from the Engineering and Physical Sciences Research Council (EP/V050516/1). B.R.B. acknowledges the Natural Environment Research Council (NE/P018459/1). Fortius Surgical Centre, Marylebone, London is acknowledged for the generous provision of space to conduct the measurements. Vyaire[TM] Medical is acknowledged for the loan of CPET testing equipment. We acknowledge the participants and players of Harlequins Rugby Club for their participation in this study. K.E.J.P. was supported by the Imperial College Clinician Investigator Scholarship (no specific grant number/code).

## Author contributions
C.M.O., P.L.S., N.A.W., J.P.R and D.C. led the study design and securing funding. C.M.O., H.E.S., B.M., J.A., K.E.J.P., S.S., J.H.H., J.P.R. and B.R.B. collected the data. C.M.O., N.A.W. and P.L.S. secured ethical approval. C.M.O., N.A.W., K.E.J.P., D.C. and J.C. managed the registration and coordination of participant volunteers and secured access to the operating theatres. C.M.O., H.E.S., B.M. and J.A. analysed the data. W.J.B. undertook the statistical analysis. J.H.H., J.P.R., D.C., J.C. and B.R.B. provided technical guidance and advice. C.M.O., H.E.S., B.M., N.A.W., K.E.J.P., J.H.H., B.R.B., B.S.K., J.P.R. and P.L.S. draughted the manuscript. All authors read and approved the final manuscript.

## Competing interests
The authors declare no competing interests.
