## [Peer Review File · Communications Medicine]

Reviewers' comments:

Reviewer #1 (Remarks to the Author):

Overall, this appears to be a high-quality study investigating an important topic relevant to respiratory disease transmission in different settings. The methodological and statistical analysis are appropriate, and the experiments well-conceived. I find this work appropriate for Communications Medicine and recommend publication once the authors address the comments below.

Title: Both in the title and the abstract the authors highlight the “mass emission.” Yet, the manuscript gives pretty equal discussion of particle number and particle mass emission. The decision to highlight “mass” in the title and abstract is not clear to me. I suggest that the authors generalize to “a comparison of aerosol emission” or better yet “a comparison of respiratory particle emission” as this would make clear that the paper deals with respiratory particles (or aerosols, if the authors prefer).

L73: I suggest the authors clarify what is meant by “similar aerodynamic behavior.” There are substantial differences in where aerosols are most likely to deposit in the respiratory system dependent on the particle size. This would be, in my opinion, be a component of “aerodynamic behavior.” If the authors are simply trying to indicate that particles as large as 100 microns do not readily sediment out of a room environment they might rephrase to state as such.

L77: The authors might also consider the results from [Alsved et al., 2020] and [Cappa et al., 2021].

Fig. 3: For the size distributions, the authors might consider whether the apparent downturn at the smallest size is real. APS instruments are notorious for having a fall off in detection sensitivity at their smallest size bin, which makes it difficult to fully trust a fit that peaks in this general range. The authors might consider also fitting their size distributions excluding the smallest size bin shown in the figure; providing results for both would add confidence to the determinations of the mode diameters.

Speaking distribution: While others have observed some evidence of a second mode for speaking (and which is very apparent for singing), the mode here is much more obvious than some studies. It would be useful if the authors discussed this to a greater extent.

L388: The authors assume a density of 1 g/cm³. This is reasonable, as some assumption must be made and 1 g/cm³ is a nice default value. However, I have some issue with the authors stating that water is the major component of the detected particles. When released they will rapidly desiccate until they reach the ambient relative humidity. The RH of the breath likely has little impact on the measured size. The authors might note this so that people do not mistakenly think that they are measuring diameters of extremely wet particles.

L198: I strongly encourage the authors to provide figures showing the volume-based size distributions to complement the number-based distributions. This is important given that the authors highlight the mass emissions.

L222: Please rephrase to state “in this size range (0.3 to 1 micron).” The collection efficiency increases again at very small diameters owing to diffusional collection.

L228: While less effective compared to mask wearing during vocalization, I suggest the authors still highlight that wearing face coverings would still lead to some reduction, just not as much. This could help avoid misunderstanding by the public who might (mis)interpret this to mean that face coverings are not useful during exercise. Further, the authors might alter this to indicate that wearing of high quality masks (e.g. FFP2) during exercise might be necessary.

L256: Would one be able to directly combine viral load and particle mass emission to deduce amount of infective material produced without knowing the relationship between viral load and transfer to the particles? One might think that this would depend on the mechanism and site of generation in the first place. For example, it is at least theoretically possible that the transfer of infective material is greater in the smaller particle mode than the larger. I'm not arguing that this is the case, but I would suggest that the authors include an appropriate caveat here.

L273: The authors might also mention here the use of face coverings.

L285: I suggest that it is more fair to state that the imposed 15 cm distance between the participant and the sampling funnel was the key factor leading to reduce emissions observed here for sampling in a funnel. Sampling from 15 cm distance is very different than previous efforts in which participants were very close to the funnel, with their nose/mouth even inside. Certainly movement of participants contributed to the lower concentrations, but the primary factor determining the lower concentrations likely resulted from the experimental design.

L292: I agree that these findings regarding higher SARS-CoV-2 in particles <5 microns are relevant to this study. However, given the notable focus on mass emissions here it seems important to consider the size of particles that carry most of the mass. This could be conveyed through showing a volume (or mass) distribution or even a cumulative distribution of volume or mass. Is most of the mass observed here below or above 5 microns? And how does this differ between exercise, breathing, and speaking? Certainly nearly all of the number are below 5 microns. If one considers the statistical nature of virus entrainment into particles [Anand and Mayya, 2020] it is not always clear that larger individual particles will carry more infective material than smaller particles, at least for those particles that do contain infective material. However, the probability that the particles do contain infective material would increase with size.

Line 262: extraneous comma after "exercise."

Alsved, M., A. Matamis, R. Bohlin, M. Richter, P. E. Bengtsson, C. J. Fraenkel, P. Medstrand, and J. Löndahl (2020), Exhaled respiratory particles during singing and talking, *Aerosol Science and Technology*, 54(11), 1245-1248, doi:10.1080/02786826.2020.1812502.

Anand, S., and Y. S. Mayya (2020), Size distribution of virus laden droplets from expiratory ejecta of infected subjects, *Scientific Reports*, 10(1), 21174, doi:10.1038/s41598-020-78110-x.

Cappa, C. D., W. D. Ristenpart, S. Barreda, N. M. Bouvier, E. Levintal, A. S. Wexler, and S. A. Roman (2021), A highly efficient cloth facemask design, *Aerosol Science and Technology*, 1-17, doi:10.1080/02786826.2021.1962795.

Reviewer #3 (Remarks to the Author):

Orton et al. have submitted an interesting manuscript comparing the aerosol concentration and mass emission during resting breathing, speaking, and exercise. Studies characterizing particle generation during various activities have grown considerably since the start of the COVID-19 pandemic due to the societal disruption that the SARS-CoV-2 virus has caused, with the important goal of better quantifying the infectious risk of common activities and their implications for infection control.

The authors recruited 25 healthy participants (13 males, 12 females) to perform a maximal, symptom limited cardiopulmonary exercise test to objectively quantify their exercise tolerance. After an hour of rest, participants completed a series of testing maneuvers (resting breathing, vocalization, vigorous and very vigorous exercise) in duplicate in a particle free environment using a modified face mask adapted to collect both ventilatory and particle concentration measurements using a very rigorous and standardized approach with particle number and mass emission rates calculated accordingly. Five subjects (20%) were unable to complete the very vigorous portion of exercise due to exhaustion. The authors achieved very nice objective separation in both subjective effort and minute ventilation between the different phases of testing, and demonstrated a progressive increase in particle numbers with increasing minute ventilation. Interestingly, speaking generated a similar number of particles to vigorous exercise but of different size distribution, with a bimodal diameters of 0.69 and 1.74 microns. Based on these findings the authors describe similar aerosol mass emission rates during vigorous exercise and speaking, and suggest that these findings imply that mitigation strategies for resting breathing and speech at rest should also be sufficient for the majority of commonly performed exercise activities.

This is a well designed study that supports and strengthens the existing literature in this area. Another recently published study is important to note and cite given its similarities in both methods and findings (<https://pubmed.ncbi.nlm.nih.gov/33957100/>). Strengths include the rigorous experimental design detailed in the appendices, and the use of a particle free space to ensure that measurements represent the activities measured, and eliminate confounding effects of background ambient particle concentrations. My primary concern with the manuscript as written is that I fear the authors somewhat overstate the clinical significance of what is, by definition, and exploratory study given the small number of exclusively healthy volunteers (i.e. p13 191-195), which should be tempered in the context of the major comments below.

Major Comments:

1. There is little discussion of the measurement device used for particle concentration collection, including the sampling rate. Although there are many highly sensitive commercially available particle counters, a considerable number are associated with substantial measurement error which should be disclosed and included as a limitation if relevant.
2. Although the experimental methods were rigorously documented to justify use of particle sampling at the level of the facemask instead of using a sampling cone 15 cm from the subject, there remains a risk of systematic measurement error given the proximity of the sampling tubing to the impellar housing. In our laboratory experience using the PTrack device, high flow rates lead to particle entrainment down the pathway of least resistance, reducing the number of small particles measured using similar diameter sampling tubing positioned at the mouth. This creates a risk for systematic underestimation of particle concentrations, especially at high flow rates and minute ventilation. Review of device manufacturer recommendations with regards to sampling location may

be sufficient to address this concern, otherwise this should be reflected as a potential limitation.

3. The bimodal diameter distribution clearly documented with speaking is an important difference that should be more clearly reflected in the discussion. The behavior of a particle cloud generated by a human being is complex, and significantly impacted by environmental factors as noted on page 15 lines 270-276. Although theoretically particles < 5 microns pose the potential for aerosolization, when exposed to a humid environment (i.e. a cloth mask saturated over time with breath condensate, for example) larger particles are more likely to “rain out” and will not pose the same level of aerosol risk. The authors clearly document a larger particle and mass emission rate of small particles with vigorous exercise, which may represent a more important infection control consideration given the shortcomings of surgical and cloth masks that they document. These are important differences in the data collected between speaking and vigorous exercise that should be more carefully addressed in the manuscript.

4. The significance of the calculated mass emission rates is also limited by the complex biophysical properties of the particle cloud, the potential variation in surroundings (i.e. walls or other physical barriers that may result in impaction or absorption of the particle cloud), and the volume of environmental air exchanges per hour – all of which are not explored here. Thus the discussion on p15 lines 248-269 likely represents an oversimplification of the engineering considerations involved, and neglects the logarithmic rate of particle clearance which makes the volume of air in the space in question rather than the concentration of particles a greater factor for most HEPA quality HVAC systems with high air exchange rates (<https://www.cdc.gov/infectioncontrol/guidelines/environmental/appendix/air.html> <https://www.ncbi.nlm.nih.gov/pmc/articles/PMC8463654/>).

5. The inability for even healthy subjects to perform high intensity exercise when masked is a consistent observation that we have seen in our laboratory as well, and is worth noting as a real world limitation to this data with significant implications for sports performance. This also underlines the important limitation that all testing in this study was performed in healthy subjects. Individuals with different cardiopulmonary disorders likely generate particle clouds with different characteristics, and may tolerate “standard mitigation measures” less well.

6. The manuscript is well written, but lengthy. Refocusing the sections that address the areas of concern above to identify future research questions and removing the clinical implications that remain somewhat speculative should offer opportunities to shorten the length of the manuscript without detracting from the high quality description of the methods and results.

7. It would be interesting to report how much variability was seen between subjects, and if there was any significant differences observed by gender in the supplemental materials.

We are grateful to the reviewers for their careful consideration and reviews of our manuscript. We reply to their specific comments below (our response in red) and indicate how the manuscript has been changed. Specific changes are tracked in our annotated revised manuscript.

Reviewer #1 (Remarks to the Author):

Overall, this appears to be a high-quality study investigating an important topic relevant to respiratory disease transmission in different settings. The methodological and statistical analysis are appropriate, and the experiments well-conceived. I find this work appropriate for Communications Medicine and recommend publication once the authors address the comments below.

Title: Both in the title and the abstract the authors highlight the “mass emission.” Yet, the manuscript gives pretty equal discussion of particle number and particle mass emission. The decision to highlight “mass” in the title and abstract is not clear to me. I suggest that the authors generalize to “a comparison of aerosol emission” or better yet “a comparison of respiratory particle emission” as this would make clear that the paper deals with respiratory particles (or aerosols, if the authors prefer).

We thank the reviewer for highlighting this. We have modified the title as below to reflect the overall discussion in the manuscript: *“Exercise, Speaking and Breathing at Rest: A Comparison of Respiratory Particle Emission Rates”*

L73: I suggest the authors clarify what is meant by “similar aerodynamic behavior.” There are substantial differences in where aerosols are most likely to deposit in the respiratory system dependent on the particle size. This would be, in my opinion, be a component of “aerodynamic behavior.” If the authors are simply trying to indicate that particles as large as 100 microns do not readily sediment out of a room environment, they might rephrase to state as such.

We have rephrased the sentence to include *“with respect to suspension in air”* in L72 to clarify what we meant by the aerodynamic behaviour.

L77: The authors might also consider the results from [Alsved et al., 2020] and [Cappa et al., 2021].

We thank the referee for the suggestion. We have updated the reference in L76 to include the work by Alsved et al. The work by Cappa et al. is already cited in this work.

Fig. 3: For the size distributions, the authors might consider whether the apparent downturn at the smallest size is real. APS instruments are notorious for having a falloff in detection sensitivity at their smallest size bin, which makes it difficult to fully trust a fit that peaks in this general range. The authors might consider also fitting their size distributions excluding the smallest size bin shown in the figure; providing results for both would add confidence to the determinations of the mode diameters.

The referee is correct in raising the counting efficiencies of particles measured by the APS instrument in the smallest size channels (less than 0.9 μm in aerodynamic size) and, particularly, for particles in the first size bin $< 0.523 \mu\text{m}$. The first bin (optical diameter) also has different bin width (0.25) compared to 0.03125 for all the other bins $> 0.523 \mu\text{m}$. In all data presented in this work, we do not include the smallest size bin ($< 0.523 \mu\text{m}$) in either the estimation of the number or mass concentrations, or in the reported size distributions, instead including only the 0.54 - 20 μm size range. We now make this clear in the caption of Fig. 3 and, where necessary, have corrected the main text. To check further, fits to the size distribution from 0.583 - 20 μm (right figure) are compared with Fig 3 (left) reported in the manuscript. There are no significant differences except for a minor change on breathing, where the difference can be attributed to the noisy nature of the data.

Size distributions for breathing at rest and during exercise, and while speaking. APS (0.54-20 μm left, and 0.583-20 μm right)

Speaking distribution: While others have observed some evidence of a second mode for speaking (and which is very apparent for singing), the mode here is much more obvious than some studies. It would be useful if the authors discussed this to a greater extent.

The speaking distribution is not so important for the study reported here, in which we intend to focus on breathing at rest and during exercise. However, we have discussed the distributions we report from speaking from a wider cohort of participants in greater detail in our previously published paper (see ref [28]). We additionally compare the distributions we report from this study (see Supplementary Figure 3) which are relatively consistent with previous.

L388: The authors assume a density of 1 g/cm³. This is reasonable, as some assumption must be made and 1 g/cm³ is a nice default value. However, I have some issue with the authors stating that water is the major component of the detected particles. When released they will rapidly desiccate until they reach the ambient relative humidity. The RH of the breath likely has little impact on the measured size. The authors might note this so that people do not mistakenly think that they are measuring diameters of extremely wet particles.

We agree with the referee that these are important points to consider. Indeed, the referee is correct on our assumptions of 1 g/cm³ and the phase of the detected particles by the APS. We have recently discussed the potential effect of evaporation of respiratory aerosols sampled by the APS through the funnel in great detail in a recent publication (see ref [45]), and include two key figures from that study below, reporting simulations of evaporation of aerosol droplets from high RH into an RH of 50% (Figures 1b and c). The size distribution can be assumed to equilibrate in <1 s for the sizes we examine in this work and so we can assume that all particles have adopted an equilibrated size and that we are not measuring partially equilibrated particles at larger sizes. However, we also show in the manuscript that the particle concentration is not diluted by room air and reflects the exhaled air, also suggesting that the RH remains high in the sampling funnel and into the APS. As a consequence, although we cannot be certain of the RH in the APS detection volume, we do believe that it remains high and that the particles remain solution droplets. In the absence of additional information on the sampling RH, which would be extremely challenging to measure, we suggest that our assumptions are reasonable provided they are clearly stated.

We have therefore included these sentences to the main text in the manuscript (L405):

“Additionally, our recent work based on analysis of the sampling of aerosols through the collection funnel and into the APS [45] show that the full size distributions (0.54-20 μm) reported here can fully equilibrate in size to the sampling relative humidity (RH), with sufficient time from exhalation to size measurement by the APS instrument. Although our previous study suggests the RH for the measured size distributions remains high, we cannot unambiguously state the RH at which our size distributions are measured, and this will be the subject of a future study.”

We have recently discussed the potential effect of evaporation of respiratory aerosols sampled by the APS through the funnel in great detail in a recent publication (see ref [45]), and include two key figures from that study below:

Figure 1: (b) Simulated evaporation profiles of saliva droplets from size ranging from 0.5 – 20 μm based on hygroscopic properties of saliva droplets, in conditions of 293 K and 50% RH. (c) The size distribution of aerosol particles generated by a cough at the source, reported by Johnson et al. at different transit times (see ref 45)]

L198: I strongly encourage the authors to provide figures showing the volume-based size distributions to complement the number-based distributions. This is important given that the authors highlight the mass emissions.

At the reviewer's suggestion, we have included the below volume-based size distributions in the SI (Supplementary Figure 5, copied below). As noticed by the referee in previous comments the volume size distribution for speaking at 70-80 dBA clearly show and highlight the importance of the laryngeal mode from speech aerosols. We have stressed this in the main text.

Figure 2 (a) Aerosol volume size distributions for breathing at rest and during exercise, and while speaking. Insert: Logarithmic vertical scale of (a) for clarity. Corresponding cumulative volume fraction for the activities is reported in (b).

L222: Please rephrase to state “in this size range (0.3 to 1 micron).” The collection efficiency increases again at very small diameters owing to diffusional collection.

We thank the referee for the suggestion. We have made the change to include the size range from (0.3 to 1 micron) (see Line 227)

L228: While less effective compared to mask wearing during vocalization, I suggest the authors still highlight that wearing face coverings would still lead to some reduction, just not as much. This could help avoid misunderstanding by the public who might (mis)interpret this to mean that face coverings are not useful during exercise. Further, the authors might alter this to indicate that wearing of high-quality masks (e.g. FFP2) during exercise might be necessary.

We agree with the reviewer and have rephrased L228 to include the suggestions on the wearing of face coverings (see L233).

“However, wearing high grade face coverings will still reduce aerosol emissions.”

L256: Would one be able to directly combine viral load and particle mass emission to deduce amount of infective material produced without knowing the relationship between viral load and transfer to the particles? One might think that this would depend on the mechanism and site of generation in the first place. For example, it is at least theoretically possible that the transfer of infective material is greater in the smaller particle mode than the larger. I’m not arguing that this is the case, but I would suggest that the authors include an appropriate caveat here.

Although we do infer that this is possible in the manuscript, strictly, the reviewer is correct as the efficiency of transfer of viral load into particles of different size is not known. We added the beneath sentences to the manuscript to clarify this topic. (See L262)

“However, in the absence of detailed knowledge of the variation in viral load with particle size, we consider that a mass scaling in viral load may be reasonable for particles across both the bronchiolar mode (from breathing, the smallest particle mode) and the laryngeal mode (dominated by particles in the 800 nm to 5 µm range). For larger particles arising from the oral mode, assuming a continued increase in viral load for particles of increasing size, given their very different site of origin, may become increasingly problematic”.

L273: The authors might also mention here the use of face coverings.

We have included the wearing of face coverings as a factor that can influence transmission for airborne pathogens and thank the referee for the suggestion.

L285: I suggest that it is more fair to state that the imposed 15 cm distance between the participant and the sampling funnel was the key factor leading to reduce emissions observed here for sampling in a funnel. Sampling from 15 cm distance is very different than previous efforts in which participants were very close to the funnel, with their nose/mouth even inside. Certainly movement of participants contributed to the lower concentrations, but the primary factor determining the lower concentrations likely resulted from the experimental design.

On reflection, we agree with the referee that this is likely to be a fair assessment. We now have added a statement to say (L296): “It is likely that the large distance between the participant and the funnel is a key factor in the reduced particle concentration recorded.”

L292: I agree that these findings regarding higher SARS-CoV-2 in particles <5 microns are relevant to this study. However, given the notable focus on mass emissions here it seems important to consider the size of particles that carry most of the mass. This could be conveyed through showing a volume (or mass) distribution or even a cumulative distribution of volume or mass. Is most of the mass observed here below or above 5 microns? And how does this differ between exercise, breathing, and speaking? Certainly nearly all of the number are below 5 microns. If one considers the statistical nature of virus entrainment into particles [Anand and Mayya, 2020] it is not always clear that larger individual particles will carry more infective material than smaller particles, at least for those particles that do contain infective material. However, the probability that the particles do contain infective material would increase with size.

We have addressed this in a previous comment. Indeed, the volume size distribution as well as the cumulative distribution of volume show that most (> 99 %) of the volume (mass) of the particles observed are below 5 µm across all activities. See responses to earlier comments.

Line 262: extraneous comma after “exercise.”

We thank the reviewer. This has been removed.

Reviewer #3 (Remarks to the Author):

Orton et al. have submitted an interesting manuscript comparing the aerosol concentration and mass emission during resting breathing, speaking, and exercise. Studies characterizing particle generation during various activities have grown considerably since the start of the COVID-19 pandemic due to the societal disruption that the SARS-CoV-2 virus has caused, with the important goal of better quantifying the infectious risk of common activities and their implications for infection control.

The authors recruited 25 healthy participants (13 males, 12 females) to perform a maximal, symptom limited

cardiopulmonary exercise test to objectively quantify their exercise tolerance. After an hour of rest, participants completed a series of testing maneuvers (resting breathing, vocalization, vigorous and very vigorous exercise) in duplicate in a particle free environment using a modified face mask adapted to collect both ventilatory and particle concentration measurements using a very rigorous and standardized approach with particle number and mass emission rates calculated accordingly. Five subjects (20%) were unable to complete the very vigorous portion of exercise due to exhaustion. The authors achieved very nice objective separation in both subjective effort and minute ventilation between the different phases of testing, and demonstrated a progressive increase in particle numbers with increasing minute ventilation. Interestingly, speaking generated a similar number of particles to vigorous exercise but of different size distribution, with a bimodal diameters of 0.69 and 1.74 microns. Based on these findings the authors describe similar aerosol mass emission rates during vigorous exercise and speaking, and suggest that these findings imply that mitigation strategies for resting breathing and speech at rest should also be sufficient for the majority of commonly performed exercise activities.

This is a well designed study that supports and strengthens the existing literature in this area. Another recently published study is important to note and cite given its similarities in both methods and findings (<https://pubmed.ncbi.nlm.nih.gov/33957100/>). Strengths include the rigorous experimental design detailed in the appendices, and the use of a particle free space to ensure that measurements represent the activities measured, and eliminate confounding effects of background ambient particle concentrations.

We have now included this reference in line 83 and as [ref 33]

My primary concern with the manuscript as written is that I fear the authors somewhat overstate the clinical significance of what is, by definition, and exploratory study given the small number of exclusively healthy volunteers (i.e. p13 191-195), which should be tempered in the context of the major comments below.

We understand the concerns of the reviewer. It is of course only possible to make measurements on healthy participants in this way and we add further details in our response to comment 5 below.

Major Comments:

1. There is little discussion of the measurement device used for particle concentration collection, including the sampling rate. Although there are many highly sensitive commercially available particle counters, a considerable number are associated with substantial measurement error which should be disclosed and included as a limitation if relevant.

We used an aerodynamic particle sizer from TSI with aerosol sampling rate of 1 sec. The APS has 1 L min⁻¹ sample flow rate, and 4 L min⁻¹ sheath flow rate. These flow rates are already included in the manuscript, see “*Aerosol Measurements (0.54 – 20 μm Diameter)*”. The 1-second sampling interval has also been added to the same section (L374).

We have discussed the measurement device, sampling rate, measurement errors etc. in some detail in our recently published paper [see ref 45] and have now cited this reference stating on L586 “*We have provided a comprehensive account of the challenges of making measurements of respiratory aerosol using the approaches described here, along with measurement uncertainties and limitations, in a recent publication [45].*”

2. Although the experimental methods were rigorously documented to justify use of particle sampling at the level of the facemask instead of using a sampling cone 15 cm from the subject, there remains a risk of systematic measurement error given the proximity of the sampling tubing to the impeller housing. In our laboratory experience using the PTrack device, high flow rates lead to particle entrainment down the pathway of least resistance, reducing the number of small particles measured using similar diameter sampling tubing positioned at the mouth. This creates a risk for systematic underestimation of particle concentrations, especially at high flow rates and minute ventilation. Review of device manufacturer recommendations with regards to sampling location may be sufficient to address this concern, otherwise this should be reflected as a potential limitation.

We have explored the size-dependent sampling efficiencies with varying instrument flow rate and the sampling line losses in our previous publications [28,45]. Here, we have compared the reported number concentrations and size distributions recorded when sampling from within the face mask, without face mask, and post-impeller to ensure that the measurement technique is robust. Indeed,

because of the losses we observe after the impeller, we only report and compare data recorded from within the face mask. *Most importantly, we have shown data recorded from within the mask are consistent with our previous measurements of participants speaking and breathing directly into the funnel without the complications of a face mask or impeller, and we have discussed together with comparison to our earlier measurement in greater detail in Appendix 1: Aerosol Sampling Methodology. Page 24* As a consequence, we are confident that we do not see a reduction in the reported concentration suggested by the referee. This may be a consequence of the higher flow rate (5L/min) drawn from the mask by the APS than the PTrack (1L/min).

3. The bimodal diameter distribution clearly documented with speaking is an important difference that should be more clearly reflected in the discussion. The behavior of a particle cloud generated by a human being is complex, and significantly impacted by environmental factors as noted on page 15 lines 270-276. Although theoretically particles < 5 microns pose the potential for aerosolization, when exposed to a humid environment (i.e. a cloth mask saturated over time with breath condensate, for example) larger particles are more likely to “rain out” and will not pose the same level of aerosol risk. *The authors clearly document a larger particle and mass emission rate of small particles with vigorous exercise, which may represent a more important infection control consideration given the shortcomings of surgical and cloth masks that they document.* These are important differences in the data collected between speaking and vigorous exercise that should be more carefully addressed in the manuscript.

As discussed above in response to the comment made by reviewer 1 on the speaking distribution (top of page 2 above), we have discussed the vocalisation mode in detail in previous work and refer the reviewer to that work. This manuscript is intended to focus more closely on breathing at rest and during exercise. For all of the particle sizes reported in this work (breathing and speaking), the sizes remain very low (even if they become intensely humidified and grow) and sedimentation loss rates can be largely ignored, particularly when compared to particles larger than 100 µm diameter. Indeed, we have explored this in detail in a previous publication [see Walker J. S, *et al*: *ACS Cent. Sci.* 2021, 7, 1, 200–209].

As highlighted in italics above, we believe the reviewer wishes us to clearly identify the challenges of removing aerosol sizes pre-dominantly associated with breathing (both at rest and by exercise) than speaking using face coverings. We agree with this comment wholeheartedly, and this is also consistent with some of the discussions raised by reviewer 1, and the addition of particle volume distributions and cumulative volume plots, now included in the SI. We believe the discussion on page 14, line 225 to 242 should now address the reviewers’ comment.

4. The significance of the calculated mass emission rates is also limited by the complex biophysical properties of the particle cloud, the potential variation in surroundings (i.e. walls or other physical barriers that may result in impaction or absorption of the particle cloud), and the volume of environmental air exchanges per hour – all of which are not explored here. Thus the discussion on p15 lines 248-269 likely represents an oversimplification of the engineering considerations involved, and neglects the logarithmic rate of particle clearance which makes the volume of air in the space in question rather than the concentration of particles a greater factor for most HEPA quality HVAC systems with high air exchange rates (<https://www.cdc.gov/infectioncontrol/guidelines/environmental/appendix/air.html> <https://www.ncbi.nlm.nih.gov/pmc/articles/PMC8463654/>).

We thank the referee and agree with the comment on the complex biophysical properties of respiratory particles and the potential engineering considerations to reduce exposure. However, we have always chosen to investigate the *source strength* in all of our studies (see references 28, 29 and 45 cited in the paper), along with this one, rather than to examine what happens to the aerosol cloud during dispersion from source. Such data are crucial (and challenging to measure) for subsequent modelling studies of aerosol transmission. Indeed, given the very low concentrations of respiratory aerosol generated, a quantitative measurement of respiratory aerosol at any other distance than close to source is near to impossible.

5. The inability for even healthy subjects to perform high intensity exercise when masked is a consistent observation that we have seen in our laboratory as well, and is worth noting as a real world limitation to this data with significant implications for sports performance. This also underlines the important limitation that all testing in this study was performed in healthy subjects. Individuals with different cardiopulmonary

disorders likely generate particle clouds with different characteristics, and may tolerate “standard mitigation measures” less well.

We are grateful for the suggestion that the wearing of a face covering may limit the ability of the subject to perform exercise. However, the cardiopulmonary exercise testing mask utilised in the study is routinely used and validated, in clinical and research exercise testing (recognised to add negligible inspiratory/expiratory resistance), and so the very high intensity of the exercise being tested (Very vigorous work rate = 30% of the difference between the anaerobic threshold and the maximal work load achieved in the first ramped test), is likely to be the primary factor contributing to the subjects’ inability to complete the final phase of the testing regimen. This conclusion is supported by the heartrate data presented in the manuscript. The reviewer is correct in stating that only healthy volunteers were recruited for the studies described, which was a pragmatic decision made around the relatively limited sample size of n= 25. We agree that individuals with differing cardiopulmonary (and other) disorders may behave differently but consider the study of healthy participants necessary to be completed first (L307).

6. The manuscript is well written, but lengthy. Refocusing the sections that address the areas of concern above to identify future research questions and removing the clinical implications that remain somewhat speculative should offer opportunities to shorten the length of the manuscript without detracting from the high quality description of the methods and results.

The length of the manuscript is within the guidance of the journal. It is not clear to us what the speculative clinical implications are that the reviewer is referring to. If this is primarily the use of face coverings, we have adapted the manuscript in response to the comments made by reviewer 1 and we believe these improve the clarity and robustness of the analysis. We have limited our discussion about the relevant size ranges for viral transmission and implications for engineering solutions as we do not believe we can derive clinical implications in these important areas.

7. It would be interesting to report how much variability was seen between subjects, and if there was any significant differences observed by gender in the supplemental materials.

We thank the referee for the suggestion. Our comparison between males (13) and females (12) across the cohorts showed no statistically significant differences in number and mass concentrations as well as emission rates (allowing for multiple comparisons). We have now included the figures below in the SI Figure 8 to address this comment.

Figure 3: Box and whisker plots showing (a) Number concentration, (b) Mass concentration, (c) Particle number emission rate and (d) Particle mass emission rate, for 13 males (blue), 12 females (red), respectively, across all the activities.

Reviewers' comments:

Reviewer #1 (Remarks to the Author):

First, I apologize to the authors (and editor) for the delay in my review. I believe that the authors have done a thorough job in their revisions and satisfactorily addressed most everything that was raised by both reviewers. My remaining concern has to do with the addition of the statement that "However, in the absence of detailed knowledge of the variation in viral load with particle size, we consider that a mass scaling in viral load may be reasonable for particles across both the bronchiolar mode..." What is unclear to me is what makes this a "reasonable" assumption. This would be greatly strengthened if it could be supported by appropriate references or physical arguments. I think that one could make a "reasonable" counterargument to a mass scaling assumption based on the physics of particle generation and entrainment of smaller entities such as viruses into those particles. I see their assumption really more as speculation.

Reviewer #3 (Remarks to the Author):

Orton et al. have submitted a revised manuscript describing respiratory particle and aerosol mass emission rates from 25 healthy subjects at rest, speaking, and during different intensities of exercise. The authors have thoughtfully revised their manuscript based on the reviewer comments with excellent overall result.

I remain concerned with the strength of the statements contained on page 15 lines 283-287 and page 17 lines 343-346, especially when tempered by the paragraph immediately prior and page 16 Lines 296-302. I agree that these findings are an important contribution to ongoing efforts to allow the safe practice of sports and exercise worldwide. However, given the limitations of this study (small sample size, healthy volunteers) and our current limited understanding of the complex interplay of factors that govern infective transmission (detailed on the top of page 16), these findings suggest that mitigation strategies deemed appropriate for speech and breathing at rest in the context of the pandemic may also be suitable for the majority of exercise settings. To be more conclusive I fear increases the risk of misinterpretation by the casual reader, and overly minimizes the limitations carefully articulated in the discussion.

Minor Comments

p 4 Line 72 Consider deleting the comma after behavior

Reviewers' comments:

Reviewer #1 (Remarks to the Author):

First, I apologize to the authors (and editor) for the delay in my review. I believe that the authors have done a thorough job in their revisions and satisfactorily addressed most everything that was raised by both reviewers. My remaining concern has to do with the addition of the statement that "However, in the absence of detailed knowledge of the variation in viral load with particle size, we consider that a mass scaling in viral load may be reasonable for particles across both the bronchiolar mode..." What is unclear to me is what makes this a "reasonable" assumption. This would be greatly strengthened if it could be supported by appropriate references or physical arguments. I think that one could make a "reasonable" counterargument to a mass scaling assumption based on the physics of particle generation and entrainment of smaller entities such as viruses into those particles. I see their assumption really more as speculation.

In agreement with the reviewers' comment, we have updated the text, removing the assumption and have replaced it with "However, the variation in viral load with particle size remains uncertain."

Reviewer #3 (Remarks to the Author):

Orton et al. have submitted a revised manuscript describing respiratory particle and aerosol mass emission rates from 25 healthy subjects at rest, speaking, and during different intensities of exercise. The authors have thoughtfully revised their manuscript based on the reviewer comments with excellent overall result.

I remain concerned with the strength of the statements contained on page 15 lines 283-287 and page 17 lines 343-346, especially when tempered by the paragraph immediately prior and page 16 Lines 296-302. I agree that these findings are an important contribution to ongoing efforts to allow the safe practice of sports and exercise worldwide. However, given the limitations of this study (small sample size, healthy volunteers) and our current limited understanding of the complex interplay of factors that govern infective transmission (detailed on the top of page 16), these findings suggest that mitigation strategies deemed appropriate for speech and breathing at rest in the context of the pandemic may also be suitable for the majority of exercise settings. To be more conclusive I fear increases the risk of misinterpretation by the casual reader, and overly minimizes the limitations carefully articulated in the discussion.

In agreement with the reviewers' comment, we have updated the text and abstract to read that mitigation strategies deemed appropriate for speech and breathing at rest in the context of the pandemic "may be suitable for the majority of exercise settings".

Minor Comments

p 4 Line 72 Consider deleting the comma after behavior

Removed